REPRODUCIBILITY IN CANCER BIOLOGY

# Challenges for assessing replicability in preclinical cancer biology

**Abstract** We conducted the Reproducibility Project: Cancer Biology to investigate the replicability of preclinical research in cancer biology. The initial aim of the project was to repeat 193 experiments from 53 high-impact papers, using an approach in which the experimental protocols and plans for data analysis had to be peer reviewed and accepted for publication before experimental work could begin. However, the various barriers and challenges we encountered while designing and conducting the experiments meant that we were only able to repeat 50 experiments from 23 papers. Here we report these barriers and challenges. First, many original papers failed to report key descriptive and inferential statistics: the data needed to compute effect sizes and conduct power analyses was publicly accessible for just 4 of 193 experiments. Moreover, despite contacting the authors of the original papers, we were unable to obtain these data for 68% of the experiments. Second, none of the 193 experiments were described in sufficient detail in the original paper to enable us to design protocols to repeat the experiments, so we had to seek clarifications from the original authors. While authors were *extremely* or *very helpful* for 41% of experiments, they were *minimally helpful* for 9% of experiments, and *not at all helpful* (or did not respond to us) for 32% of experiments. Third, once experimental work started, 67% of the peer-reviewed protocols required modifications to complete the research and just 41% of those modifications could be implemented. Cumulatively, these three factors limited the number of experiments that could be repeated. This experience draws attention to a basic and fundamental concern about replication – it is hard to assess whether reported findings are credible.

**TIMOTHY M ERRINGTON\*, ALEXANDRIA DENIS[†], NICOLE PERFITO[‡], ELIZABETH IORNS, BRIAN A NOSEK**

**\*For correspondence:** tim@cos.io

**Present address:** [†]Fordham University School of Law, New York, United States; [‡]Rarebase, Palo Alto, United States

## Introduction

Science is a system for accumulating knowledge. The credibility of knowledge claims relies, in part, on the transparency and repeatability of the evidence used to support them. As a social system, science operates with norms and processes to facilitate the critical appraisal of claims, and transparency and skepticism are virtues endorsed by most scientists (*Anderson et al., 2007*). Science is also relatively non-hierarchical in that there are no official arbiters of the truth or falsity of claims. However, the interrogation of new claims and evidence by peers occurs continuously, and most formally in the peer review of manuscripts prior to publication. Once new claims are made public, other scientists may question, challenge, or extend them by trying to replicate the evidence or to conduct novel research. The evaluative processes of peer review and replication are the basis for believing that science is self-correcting. Self-correction is necessary because mistakes and false starts are expected when pushing the boundaries of knowledge. Science works because it efficiently identifies those false starts and redirects resources to new possibilities.

We believe everything we wrote in the previous paragraph except for one word in the last sentence – efficiently. Science advances knowledge and is self-correcting, but we do not believe it is doing so very efficiently. Many parts of research could improve to accelerate discovery. In this paper, we report the challenges confronted during a large-scale effort to replicate findings in cancer biology, and describe how improving transparency and sharing can make it easier to assess rigor and replicability and, therefore, to increase research efficiency.

Transparency is essential in any system that seeks to evaluate the credibility of scientific claims. To evaluate a scientific claim one needs access to the evidence supporting the claim – the methodology and materials used, the data generated, and the process of drawing conclusions from those data. The standard process for providing this information is to write a research paper that details the methodology and outcomes. However, this process is imperfect. For example, selectively reporting experiments or analyses, particularly reporting only those that 'worked', biases the literature by ignoring negative or null results (*Fanelli, 2010*; *Fanelli, 2011*; *Ioannidis, 2005*; *Rosenthal, 1979*; *Sterling, 1959*; *Sterling et al., 1995*). And the combined effect of constraints related to the research paper format (including word limits, and only reporting what can be described in words), the tendency of authors to report what they perceive to be important, and rewards for exciting, innovative outcomes is an emphasis on reporting outcomes and their implications, rather than a comprehensive description of the methodology (*Kilkenny et al., 2009*; *Landis et al., 2012*; *Moher et al., 2008*).

The sharing of data, materials, and code can also increase the efficiency of research in a number of ways (*Molloy, 2011*; *Murray-Rust et al., 2010*; *Nosek et al., 2015*). For example, sharing provides opportunities for independent observers to evaluate both the evidence reported in papers and the credibility of the claims based on this evidence; it allows other researchers to analyze the data in different ways (by, for example, using different rules for data exclusion); and it helps other researchers to perform replications to determine if similar evidence can be observed independently of the original context. Moreover, giving other researchers access to data, materials, and code may allow them to identify important features of the research that were not appreciated by the original researchers, or to identify errors in analysis or reporting.

Transparency and sharing therefore contribute to assessment of research reproducibility, robustness, and replicability. Reproducibility refers to whether the reported findings are repeatable using the same analysis on the same data as the original study. Robustness refers to whether the reported findings are repeatable using reasonable alternative analysis strategies on the same data as the original study. Replicability refers to whether the reported findings are repeatable using new data (*NAS, 2019*). By these definitions, all reported findings should be reproducible in principle; variability in robustness may imply fragility of the phenomenon or greater uncertainty in its evidence base; and variability in replicability may imply fragility, more limited scope of applicability than originally presumed, or uncertainty in the conditions necessary for observing supporting evidence (*Nosek and Errington, 2020a*). All three are important for assessing the credibility of claims and to make self-corrective processes as efficient as possible.

From 2013 to 2020, as part of the Reproducibility Project: Cancer Biology, we tried to replicate selected results in high-impact preclinical papers in the field of cancer biology (*Errington et al., 2014*; *Table 1*). The aim of the project was not to repeat every experiment in each paper: rather it was to repeat a selection of key experiments from each paper. The project also adopted an approach in which a Registered Report describing the experimental protocols and plans for data analysis had to be peer reviewed and accepted for publication before experimental work could begin. The Replication Study reporting the results of the experiments was then peer reviewed to ensure that the experiments had been conducted and analyzed according to the procedures outlined in the Registered Report: crucially, reviewers were asked not to take the 'success' or 'failure' of the experiments into account when reviewing Replication Studies.

The initial goal was to repeat 193 experiments from 53 high-impact papers published between 2010 and 2012, but the obstacles we encountered at every phase of the research lifecycle meant that we were only able to repeat 50 experiments from 23 papers. In a separate paper we report a meta-analysis of the results of those 50 experiments (*Errington et al., 2021b*). In this paper, we describe the challenges we confronted during the different phases of the research lifecycle. A completed replication attempt passed through six phases: designing the experiment (and writing the Registered Report); peer reviewing the Registered Report; preparing the experiments; conducting the experiments; analysing the data (and writing the Replication Study); and peer reviewing the Replication Study.

The next section discusses in detail the challenges faced during the first of these phases. A subsequent section covers the challenges encountered when conducting the experiments and during the peer review of the Replication Studies.

**Table 1.** The 53 papers selected for replication in the RP:CB.

| Original paper | Experiments selected | Registered report | Experiments registered | Replication study* | Experiments completed | Data, digital materials, and code |
|---|---|---|---|---|---|---|
| Poliseno et al., 2010 | 11 | Khan et al., 2015 | 6 | Kerwin et al., 2020 | 5 | https://osf.io/yyqas/ |
| Sharma et al., 2010 | 8 | Haven et al., 2016 | 8 | N/A | 0 | https://osf.io/xbign/ |
| Gupta et al., 2010 | 2 | N/A | 0 | N/A | 0 | https://osf.io/4bokd/ |
| Figueroa et al., 2010 | 6 | N/A | 0 | N/A | 0 | https://osf.io/xdojz/ |
| Ricci-Vitiani et al., 2010 | 3 | Chroscinski et al., 2015b | 2 | Errington et al., 2021a | 1 | https://osf.io/mpyvx/ |
| Kan et al., 2010 | 3 | Sharma et al., 2016a | 3 | Errington et al., 2021a | 1 | https://osf.io/jpeqg/ |
| Heidorn et al., 2010 | 8 | Bhargava et al., 2016a | 5 | Errington et al., 2021a | 1 | https://osf.io/b1aw6/ |
| Hatzivassiliou et al., 2010 | 4 | Bhargava et al., 2016b | 3 | Pelech et al., 2021 | 2 | https://osf.io/0hezb/ |
| Vermeulen et al., 2010 | 4 | Evans et al., 2015a | 3 | Essex et al., 2019 | 3 | https://osf.io/pgjhx/ |
| Carro et al., 2010 | 8 | N/A | 0 | N/A | 0 | https://osf.io/mfxpj/ |
| Nazarian et al., 2010 | 5 | N/A | 0 | N/A | 0 | https://osf.io/679uw/ |
| Johannessen et al., 2010 | 5 | Sharma et al., 2016b | 5 | Errington et al., 2021a | 2 | https://osf.io/lmhjg/ |
| Poulikakos et al., 2010 | 5 | N/A | 0 | N/A | 0 | https://osf.io/acpq7/ |
| Sugahara et al., 2010 | 4 | Kandela et al., 2015a | 3 | Mantis et al., 2017 | 3 | https://osf.io/xu1g2/ |
| Ward et al., 2010 | 3 | Fiehn et al., 2016 | 3 | Showalter et al., 2017 | 3 | https://osf.io/8l4ea/ |
| Ko et al., 2010 | 3 | N/A | 0 | N/A | 0 | https://osf.io/udw78/ |
| Zuber et al., 2011 | 3 | N/A | 0 | N/A | 0 | https://osf.io/devog/ |
| Delmore et al., 2011 | 2 | Kandela et al., 2015b | 2 | Aird et al., 2017 | 2 | https://osf.io/7zqxp/ |
| Goetz et al., 2011 | 2 | Fiering et al., 2015 | 2 | Sheen et al., 2019 | 2 | https://osf.io/7yqmp/ |
| Sirota et al., 2011 | 1 | Kandela et al., 2015c | 1 | Kandela et al., 2017 | 1 | https://osf.io/hxrmm/ |
| Raj et al., 2011 | 4 | N/A | 0 | N/A | 0 | https://osf.io/uvapt/ |
| Possemato et al., 2011 | 3 | N/A | 0 | N/A | 0 | https://osf.io/u1mfn/ |

*Table 1 continued on next page*

*Table 1 continued*

| Original paper | Experiments selected | Registered report | Experiments registered | Replication study* | Experiments completed | Data, digital materials, and code |
|---|---|---|---|---|---|---|
| *Tay et al., 2011* | 5 | *Phelps et al., 2016* | 5 | *Wang et al., 2020* | 4 | https://osf.io/oblj1/ |
| *Xu et al., 2011* | 5 | *Evans et al., 2015b* | 5 | N/A | 0 | https://osf.io/kvshc/ |
| *DeNicola et al., 2011* | 4 | N/A | 0 | N/A | 0 | https://osf.io/i0yka/ |
| *Zhu et al., 2011* | 3 | N/A | 0 | N/A | 0 | https://osf.io/oi7ji/ |
| *Liu et al., 2011* | 4 | *Li et al., 2015* | 3 | *Yan et al., 2019* | 3 | https://osf.io/gb7sr/ |
| *Dawson et al., 2011* | 3 | *Fung et al., 2015* | 3 | *Shan et al., 2017* | 3 | https://osf.io/hcqqy/ |
| *Qian et al., 2011* | 3 | N/A | 0 | N/A | 0 | https://osf.io/ckpsn/ |
| *Sumazin et al., 2011* | 3 | N/A | 0 | N/A | 0 | https://osf.io/wcasz/ |
| *Chaffer et al., 2011* | 2 | N/A | 0 | N/A | 0 | https://osf.io/u6m4z/ |
| *Opitz et al., 2011* | 5 | N/A | 0 | N/A | 0 | https://osf.io/o2xpf/ |
| *Kang et al., 2011* | 2 | *Raouf et al., 2015* | 2 | N/A | 0 | https://osf.io/82nfe/ |
| *Chen et al., 2012* | 2 | N/A | 0 | N/A | 0 | https://osf.io/egoni/ |
| *Driessens et al., 2012* | 2 | N/A | 0 | N/A | 0 | https://osf.io/znixv/ |
| *Garnett et al., 2012* | 3 | *Vanden Heuvel et al., 2016* | 3 | *Vanden Heuvel et al., 2018* | 3 | https://osf.io/nbryi/ |
| *Schepers et al., 2012* | 3 | N/A | 0 | N/A | 0 | https://osf.io/1ovqn/ |
| *Willingham et al., 2012* | 2 | *Chroscinski et al., 2015a* | 1 | *Horrigan and Reproducibility Project: Cancer Biology, 2017a* | 1 | https://osf.io/9pbos/ |
| *Straussman et al., 2012* | 4 | *Blum et al., 2014* | 4 | N/A | 0 | https://osf.io/p4lzc/ |
| *Arthur et al., 2012* | 2 | *Eaton et al., 2015* | 2 | *Eaton et al., 2018* | 2 | https://osf.io/y4tvd/ |
| *Peinado et al., 2012* | 3 | *Lesnik et al., 2016* | 2 | *Kim et al., 2018* | 2 | https://osf.io/ewqzf/ |
| *Malanchi et al., 2011* | 3 | *Incardona et al., 2015* | 2 | N/A | 0 | https://osf.io/vseix/ |
| *Berger et al., 2012* | 1 | *Chroscinski et al., 2014* | 1 | *Horrigan et al., 2017b* | 1 | https://osf.io/jvpnw/ |

*Table 1 continued on next page*

Table 1 continued

| Original paper | Experiments selected | Registered report | Experiments registered | Replication study* | Experiments completed | Data, digital materials, and code |
|---|---|---|---|---|---|---|
| **Prahallad et al., 2012** | 4 | N/A | 0 | N/A | 0 | https://osf.io/ecy85/ |
| **Wilson et al., 2012** | 3 | **Greenfield et al., 2014** | 2 | N/A | 0 | https://osf.io/h0pnz/ |
| **Lu et al., 2012** | 5 | **Richarson et al., 2016** | 3 | **Errington et al., 2021a** | 2 | https://osf.io/vfsbo/ |
| **Lin et al., 2012** | 2 | **Blum et al., 2015** | 2 | **Lewis et al., 2018** | 2 | https://osf.io/mokeb/ |
| **Lee et al., 2012** | 3 | N/A | 0 | N/A | 0 | https://osf.io/i25y8/ |
| **Castellarin et al., 2012** | 1 | **Repass et al., 2016** | 1 | ***Repass and Reproducibility Project: Cancer Biology, 2018*** | 1 | https://osf.io/v4se2/ |
| **Crasta et al., 2012** | 3 | N/A | 0 | N/A | 0 | https://osf.io/47xy6/ |
| **Png et al., 2011** | 5 | N/A | 0 | N/A | 0 | https://osf.io/tkzme/ |
| **Metallo et al., 2011** | 5 | N/A | 0 | N/A | 0 | https://osf.io/isdbh/ |
| **Morin et al., 2010** | 1 | N/A | 0 | N/A | 0 | https://osf.io/6kuy8/ |

193 experiments in 53 papers were selected for replication. The papers are listed in column 1, and the number of experiments selected from each paper is listed in column 2. Registered Reports for 87 experiments from 29 papers were published in *eLife*. The Registered Reports are listed in column 3, and the number of experiments included in each Registered Report is listed in column 4. 50 experiments from 23 Registered Reports were completed. 17 Replication Studies reporting the results of 41 experiments were published in *eLife*; the results of another nine experiments from the six remaining Registered Reports were published in an aggregate paper (**Errington et al., 2021a**). The Replication Studies are listed in column 5, and the number of experiments included in each study is listed in column 6. Column seven contains a link to data, digital materials, and code.

## The challenges encountered when designing experiments

### Sampling papers

At the start of the project in 2013 we searched various databases to identify basic research papers in cancer biology published between 2010 and 2012 that were having a substantial impact as indexed by citation rates and readership in multiple databases. We selected the highest impact papers from each year that met inclusion criteria (*Errington et al., 2014*). We excluded papers that reported exclusively genomics, proteomics, and high-throughput assays. This resulted in 50 included papers for which we initiated the process of preparing a replication. During inquiries with original authors, two papers were identified that we determined would be unfeasible to attempt and we decided to halt the effort; for another paper we requested, but did not receive, a key material (i.e., mouse model) so replication was not feasible. We decided to go back to the sampling pool and pull the next available papers, bringing the effective sample to 53 papers. Observing that challenges like this were relatively common, we did not return to the pool for resampling again for the rest of the project. Among the 53 selected papers, 35 were published in the Nature family of journals, 11 in the Cell family of journals, 4 in the Science family of journals, and three in other journals.

From each paper, we identified a subset of experiments for potential replication with an emphasis on those supporting the main conclusions of the paper and attending to resource constraints (*Table 1*). In total, 193 experiments were identified for replication across the 53 papers for an average of 3.6 per paper (SD = 1.9; range 1–11). *Figure 1* illustrates the fate of all the experiments that we attempted to replicate. Below, we summarize the findings by experiment; similar findings are observed when aggregating the data by paper (*Figure 1—figure supplement 1*).

### Searching for data from the original experiments

We planned to conduct replications with at least 0.80 power to detect the effect size reported in the original paper at $p < .05$ using two-tailed tests. However, in a number of cases only representative images or graphs were reported in the original paper. This occurred for 53 of the 193 experiments (27%). Additionally, it was uncommon for papers to include the summary statistics (such as sample size, means, standard deviations, and inferential statistics) that were needed to calculate the original effect size. We searched the original paper and supplemental files for the original data. When data were not publicly accessible, we requested them from the original authors. At least some data were open or included in the paper for four experiments (2%), raw data were shared for 31 experiments (16%), summary data were shared for 27 experiments (14%), and nothing was shared for 131 experiments (68%).

Failure to report sample size, variability information from sampling, or inferential tests in the original paper makes it difficult or impossible to calculate effect sizes. Further, failure to share data upon request – even summary statistics – leaves the nature of the original research and inference testing opaque. When we could not obtain the data we needed we estimated means and variability from the available information reported in the original papers (e.g., estimating bar heights and error bars from graphs). In cases where there was no information to estimate, such as only a representative image, we treated the extracted representative data point as the mean and estimated a range of variances to determine the replication sample size (*Errington et al., 2014*).

Analytic code availability was not common, although, unlike data, we did not explicitly request it for all experiments. Statistical analyses were reported for 78 of the 193 experiments (40%). When the outcome of analyses were reported (e.g., *p*-value) it was unclear what statistical test was used in 16 of the 78 experiments (21%). Of the experiments that reported an outcome from statistical analyses, at least some analysis code was open for one experiment (1%), code was shared by the original authors for 10 experiments (13%), additional analysis details were shared for four experiments (5%), and nothing was shared for 63 experiments (81%).

### Independent development of replication protocols

To carry out rigorous replication studies, we needed to understand the original methodology. We read each paper and supplementary information closely to design a protocol. We coded if we requested a key reagent (i.e., cell lines, plasmids, model organisms, antibodies) that was not available commercially or in a repository. We requested key reagents for 136 of the 193 experiments (70%) and for 45 of the 53 papers (85%).

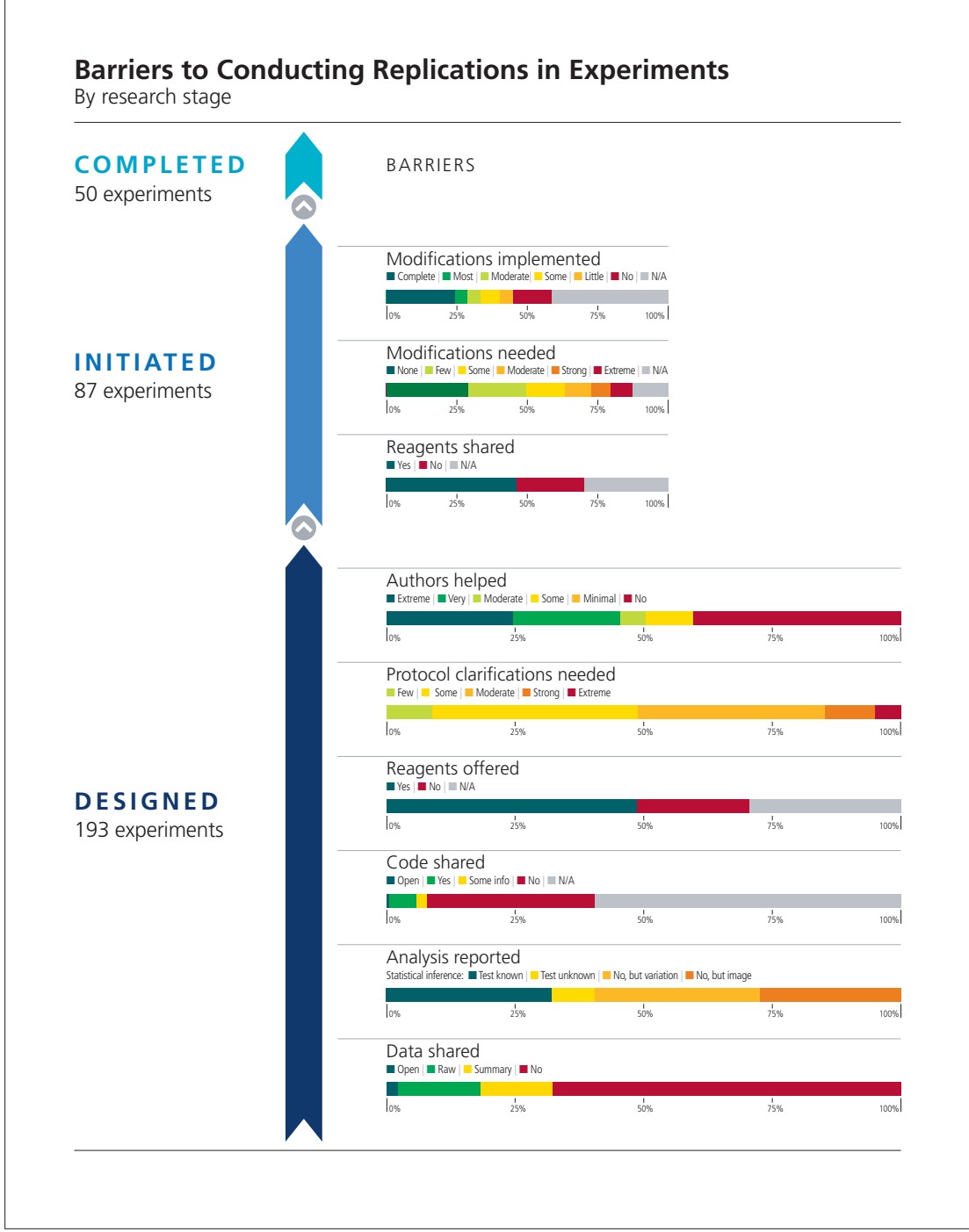

**Figure 1.** Barriers to conducting replications – by experiment. During the design phase of the project the 193 experiments selected for replication were coded according to six criteria: availability and sharing of data; reporting of statistical analysis (i.e., did the paper describe the tests used in statistical analysis?; if such tests were not used, did the paper report on biological variation (e.g., graph reporting error bars) or representative images?); availability and sharing of analytic code; did the original authors offer to share key reagents?; what level of protocol clarifications were needed from the original authors?; how helpful were the responses to those requests? The 29 Registered Reports published by the project included protocols for 87 experiments, and these experiments were coded according to three criteria: were reagents shared by the original authors?; did the replication authors have to make modifications to the protocol?; were these modifications implemented? A total of 50 experiments were completed.

The online version of this article includes the following figure supplement(s) for figure 1:

**Figure supplement 1.** Barriers to conducting replications – by paper.

> ## Box 1. Case study: Designing a replication protocol by reading the original paper.
>
> Designing the replication protocol (*Kandela et al., 2015a*) for measuring the effect of doxorubicin alone or in combination with a tumor penetrating peptide in mice bearing orthotopic prostate tumors was challenged by a lack of details in the original paper (*Sugahara et al., 2010*). There was no detailed protocol for the peptide generation in the paper or the cited references. Instead, the sequence and a general description of the 'standard' technique was briefly described. Data variability, sample size, and statistical analyses were reported; however, no raw data was available. The strain and sex of the mice and the cell type and number of cells implanted were provided; however, there were no detailed protocols available for generating or harvesting the orthotopic prostate tumors which meant these details were filled in based on the standard approach used in the replicating laboratory. Most end-point measurements were described or discernable; however, there was no description of how 'positive area' was determined for TUNEL staining which meant this needed to be surmised and articulated for the replication attempt. This paper was coded as no data available beyond what was reported in graphs and images in the original paper, statistical analysis reported with tests described with no code available beyond the reported analysis, and "strong clarification" needed about the published experimental methodology.

We coded the frequency with which we were able to design a complete protocol for repeating an experiment based on the original paper without having to contact the original authors to clarify some aspect of the original experiment (see Case study in *Box 1*). Zero experiments needed no clarifications (0%), 17 experiments needed few clarifications (9%), 77 experiments needed some clarifications (40%), 60 experiments needed moderate clarifications (31%), 29 experiments needed strong clarifications (15%), 10 experiments needed extreme clarifications (5%). To illustrate, one experiment needing few clarifications was missing reagent identifying information (e.g., catalog numbers), cell density at time of transfection (or harvest), and some specific details about the gas chromatography-mass spectrometry methodology (e.g., ramping, derivatization volume, injection volume). An experiment needing moderate clarifications was missing reagent identifying information, specific details about the transfection and infection methodologies (e.g., cell density, amount of plasmid/viral titer), and specific details about the flow cytometry methodology (e.g., cell dissociation technique, specific gating strategy). And, an experiment needing extreme clarifications was missing reagent identifying information, specific details about the transfection and infection methodologies, specific details for injecting mice with cells (e.g., number of cells and volume injected, injection methodology), specific details about the bioluminescence imaging (e.g., amount and location of luciferin injected, time post-injection until measurement), and clarification of measurement details (e.g., the exact days post-injection when measurements were taken, how the reported ratio values were calculated).

### Requesting assistance from original authors

We sought assistance from original authors to clarify the experimental protocols and to obtain original materials and reagents when necessary. We sent authors the drafted experimental protocols, clarification questions, and requests for materials. Some original authors were helpful and generous with their time providing feedback (see Case study in *Box 2*), others were not. We coded if original authors were willing to share key reagents. Of the 45 papers for which we requested a key reagent, the authors of 33 papers (73%) offered to share at least one key material. By experiment, of the 136 experiments for which we requested a key reagent, the authors were willing to share for 94 of them (69%).

We also coded the degree to which authors were helpful in providing feedback and materials for designing the replication experiments. Authors were extremely helpful for 51 experiments (26%), very helpful for 28 experiments (15%), moderately helpful for 18 experiments (9%), somewhat helpful for 18 experiments (9%), minimally helpful for 17 experiments (9%), and not at all helpful/no response for 61 experiments (32%). An example of an extremely helpful response was the

## Box 2. Case study: Feedback from original authors.

The replication protocol (*Fiering et al., 2015*) for evaluating the impact stromal caveolin-1 has on remodeling the intratumoral microenvironment was challenged by a lack of details in the original paper (*Goetz et al., 2011*). However, the original authors supplied us with most of the missing details. Based on the description in the paper, multiple strains of knockout mice could have been used for the replications. The authors provided strain stock numbers ensuring the same genetic background was selected. The authors also shared the raw data and statistical analysis: this was particularly helpful for understanding the original effects and sample size planning because the data did not have a normal distribution. The tumor cells used in the original study, engineered to express luciferase, were not available in a repository but the original authors provided them upon request. The authors also provided detailed protocol information and clarified uncertainties with reporting in the original paper. This included the age of the mice, injection details of the cells and luciferin (e.g., location, timing, procedural details), a detailed immunostaining and microscopy protocol (e.g., number of fields taken per section, magnification and specs of the objective), and the euthanasia criteria that was approved by the original study's ethics committee. The latter determined the number of days the mice were maintained in the original study. The authors also shared an additional assay, which was included in the published Replication Study (*Sheen et al., 2019*), that demonstrated the extracellular matrix remodeling capabilities of the cells that was not shown in the original paper because journal policy restricted the number of supplemental figures. This paper was coded as raw data shared by original authors, statistical analysis reported with tests described and code shared by original authors, original authors offered to share key reagents, and "extremely helpful" response from the original authors to the "moderate clarification" needed about the published experimental methodology.

corresponding author reaching out to the other authors (who since moved to other institutions) to help with the requests, sharing detailed protocol and reagent information, providing additional information beyond what we requested to help ensure the experimental details were complete, and providing additional feedback on any known deviations that were needed (e.g., different instrumentation) to help ensure a good-faith replication would be designed. An example of a moderately helpful response was replying to all of our requests with the necessary information and providing additional clarifications when follow-up requests were made, but where some parts of the response were not very helpful. For example, a request for specific protocol details was responded with "a standard procedure was used." Examples of not at all helpful responses include non-response to multiple requests (6/53 papers [11%]) or responses questioning the value of conducting replications and declining to assist.

An obvious hypothesis is that the helpfulness of the original authors was determined by the extent of clarifications requested because of the workload. If only minimal clarification were needed, then authors would be helpful. If lots of clarifications were needed, then authors would

not be helpful. The correlation between extent of clarifications and helpfulness was –0.24 (95% CI [–0.48, 0.03]) across papers and –0.20 (95% CI [–0.33, –0.06]) across experiments. Larger requests were only modestly associated with less helpfulness. The variability in this relationship is visualized in *Figure 2*. We also explored whether the extent of clarifications or helpfulness varied by experimental techniques and found the relationship was similar across different categories of experimental techniques (*Figure 2—figure supplement 1*; *Figure 2—figure supplement 2*).

### Preparing the Registered Report for peer review

Depending on feedback and materials received from original authors, some protocols were easier to design than others. To design experiments with at least 0.80 power to detect the effect size reported in the original paper at $p < .05$ using two-tailed tests, we often needed a larger sample size for the replication than what was reported in the original experiment. As an illustration, the average sample size of animal experiments in the replication protocols (average = 30; SD = 16; median = 26; IQR = 18–41) were 25% higher than the sample size of the original

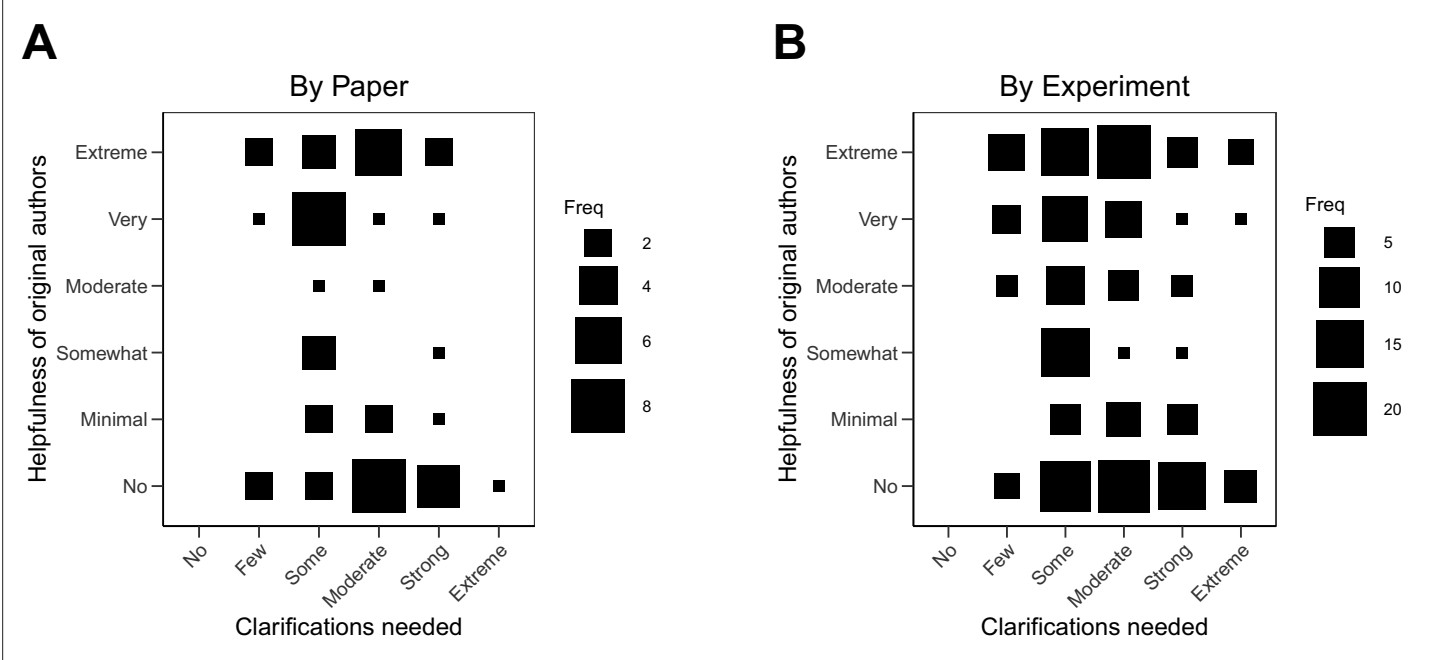

**Figure 2.** Relationship between extent of clarification needed and helpfulness of authors. Fluctuation plots showing the coded ratings for extent of clarifications needed from original authors and the degree to which authors were helpful in providing feedback and materials for designing the replication experiments. The size of the square shows the number (Freq) of papers/experiments for each combination of extent of clarification needed and helpfulness. (**A**) To characterize papers (N = 53), coded ratings were averaged across experiments for each paper. The average number of experiments per paper was 3.6 (SD = 1.9; range = 1–11). The Spearman rank-order correlation between extent of clarification needed and helpfulness was –0.24 (95% CI [–0.48, 0.03]) across papers. (**B**) For experiments (N = 193), the Spearman rank-order correlation between extent of clarification needed and helpfulness was –0.20 (95% CI [–0.33, –0.06]).

The online version of this article includes the following figure supplement(s) for figure 2:

**Figure supplement 1.** Techniques used in the original experiments.

**Figure supplement 2.** Relationship between extent of clarification needed or helpfulness with category of techniques.

---

## Box 3. Case study: Gathering original materials.

The replication protocols (**Khan et al., 2015**) for evaluating the impact of *PTENP1* on cellular PTEN expression and function required a plasmid that overexpressed the 3'UTR of *PTEN*. The original paper (**Poliseno et al., 2010**) described the generation of this plasmid and the original authors agreed to share this plasmid, as indicated in the Registered Report (**Khan et al., 2015**). A material transfer agreement (MTA) was initiated to obtain the plasmid. More than one year passed without the MTA being finalized preventing us from acquiring the plasmid. To complete the replication study, we regenerated the plasmid adding time and cost to the study. The regenerated plasmid for the replication study was deposited in Addgene (plasmid# 97204; RRID:Addgene_97204) for the research community to easily access for future research. These experiments were coded as key reagents offered to be shared, but not actually shared.

experiments (average = 24; SD = 14; median = 22; IQR = 16–30). Also, some experiments proved challenging to complete, or were discontinued, due to delays and cost increases that emerged when the replications were being designed and/or conducted (e.g., when the original authors declined to share reagents, or it became clear that the material transfer agreement process was going to take a very long time (see Case study in **Box 3**)). This included discontinuing some viable experiments that were still near the start of the design phase to ensure that experiments

## Box 4. Case study: Solving challenges during data collection.

The replication protocol (*Kandela et al., 2015b*) for evaluating BET bromodomain inhibition as a therapeutic strategy to target c-Myc described the timeframe for tumor cell inoculation, injection with luciferin to image the tumor progression, and injection with a BET bromodomain inhibitor. This followed the same timing as the original study (*Delmore et al., 2011*). An initial attempt was unsuccessful in detecting bioluminescence even though disease progression was observed, indicating tumor cell inoculation occurred, and the luciferase expressing tumor cells had a strong luminescent signal prior to injection. Lack of bioluminescence meant that we could not test the BET bromodomain inhibitor because the predetermined baseline bioluminescence indicating disease progression for inhibitor administration was never achieved. We modified the preregistered protocol and selected for highly expressing cells to enrich the tumor cells. We also designed a pilot study to identify a modified time frame in which mice could establish the same detectable baseline bioluminescence as the original study before administration of the inhibitor. We included the initial preregistered study, the pilot, and the modified replication in the published Replication Study (*Aird et al., 2017*) and discussion of the variability of the timing from tumor cell inoculation until baseline disease detection, comparing the original study, the replication, and other published studies using the same model. This experiment was coded as "completely implemented" for the "extreme modifications" needed for the experiment.

that were further along in the process could be completed.

Ultimately, 32 Registered Reports covering 97 experiments were submitted for peer review. One or more authors from the original paper was always invited by *eLife* to participate in the peer-review process. None of the papers were accepted without revision, one was not resubmitted after resource consideration of requested revisions, and two were rejected. As such, 29 papers with 87 experiments were published as Registered Reports (*Table 1*). We will now discuss some of the problems and challenges encountered in subsequent phases of the project.

### Challenges during experiments and peer review

***The challenges encountered when conducting experiments***

Once accepted as Registered Reports, experiments could begin in the replication labs. Despite often obtaining original materials and reagents and having fully specified and peer reviewed protocols, it was common that the preregistered procedure had to be modified to complete the experiments (see Case study in *Box 4*). Sometimes just minor modifications were needed (e.g., changing antibody concentrations or blocking reagents to detect the protein of interest during

a Western blot assay). Sometimes moderate modifications were needed. In some cases, for example, despite attempts to adjust the conditions, we were still unable to produce the expected intermediate results (e.g., obtaining the desired transfection efficiency as indicated by a reporter system) and an additional protocol step, different reagent source, or change in instrumentation was needed (e.g., including an enrichment step, such as fluorescence-activated cell sorting [FACS], to increase the number of transfected cells). And in some cases extreme modifications to the preregistered procedure were needed. For example, in one case (*Yan et al., 2019*) the preregistered protocol did not result in the generation of tumors in mice that were needed for a downstream infection and tumorigenicity assay, so substantial changes had to be made to this protocol to proceed with the experiment (e.g., changing the source of the tumor cells, modifying the timing and technique of infection to achieve the desired transduction efficiency, and using a different technique to detect the molecule of interest).

We coded each experiment on the extent to which modifications were needed to conduct a fair replication of the original findings. No modifications were required for 25 of the 87 experiments (29%), few modifications for 18 (21%), some modifications for 12 (14%), moderate

modifications for 8 (9%), strong modifications for 6 (7%), and extreme modifications for 7 (8%). We did not start 11 experiments and thus did not assess the level of modification required for these. This means that a total of 76 experiments were started.

The implementation of the modifications varied. When modifications could be carried out, in some cases they were completely implemented (see Case study in **Box 4**) and in others they were only partially implemented. For example, modifications were successfully implemented to reach some preregistered end-point measurements, but not all (e.g., modifications were implemented to enable quantification of one protein of interest, while continued challenges detecting another protein of interest was eventually halted). Not all modifications could be carried out. In some cases this was due to feasibility or resource constraints; and in other cases it was due to pronounced differences in the behavior of model systems or experimental protocols from what was reported in the original paper that had no obvious strategy for modification (see Case study in **Box 5**). We coded the extent to which we were able to implement the needed modifications to complete the replication experiments. Modifications were not needed for 25 of the 87 experiments (29%). We completely implemented modifications for 21 experiments (24%), mostly implemented them for four experiments (5%), moderately implemented them for four experiments (5%), implemented some of them for six experiments (7%), implemented few of them for four experiments (5%), and did not implement

any for 12 experiments (14%). As before, the 11 experiments that were not started were not assessed. Excluding papers that needed no modifications or were not assessed, the correlation between extent of modification needed and implementation of modifications was –0.01 (95% CI [–0.42, 0.40]) across papers and 0.01 (95% CI [–0.27, 0.28]) across all experiments (**Figure 3**).

Having original materials, a fully specified protocol, and peer review from experts was not always sufficient to ensure that the replication protocol behaved as expected to test the original claim. The observed implementation challenges could mean that the original finding is not valid because the model system or other parts of the protocol do not operate that way in reality – for example, the original procedure was confounded or influenced by some unrecognized factors. It might also be that, in some cases, a failure to replicate was caused by the replication team deviating from the protocol in some way that was not recognized, or that a key part of the procedure was left out of the protocol inadvertently. It is also possible that the effect reported in the original paper depended on methodological factors that were not identified by original authors, the replication team, or any other experts involved in the peer review of the original paper or the Registered Report.

Whatever the reason, all of these factors are barriers to replicability and causes of friction in efficiency of replication and discovery. Failures during implementation leave untested the original claim because the original experiment could not be carried out as described. That does not

---

## Box 5. Case study: Failing to solve challenges during data collection.

The replication protocol (*Chroscinski et al., 2015b*) for evaluating whether glioblastoma stem-like cell-derived endothelial cells contribute to tumor growth in vivo required generating cancer cells that stably expressed the thymidine kinase gene under the control of the transcriptional regulatory elements of the endothelial marker *Tie2*. The preregistered protocol required achieving at least 80% positive expression of the gene, based on a GFP reporter, among the cell populations before proceeding with the xenograft experiment. However, after multiple attempts the required expression level could not be achieved despite obtaining new cells, plasmids, and incorporating changes to the protocol suggested by the original authors to improve the infection efficiency and enrich GFP expressing cells. Eventually, the replication attempt was stopped because of the increasing costs associated with multiple optimization attempts with no feasible path to a solution. The original finding (*Ricci-Vitiani et al., 2010*), that selective killing of tumor-derived endothelial cells results in tumor reduction and degeneration, was not tested. This experiment was coded as "some implemented" for the "moderate modifications" needed for the experiment.

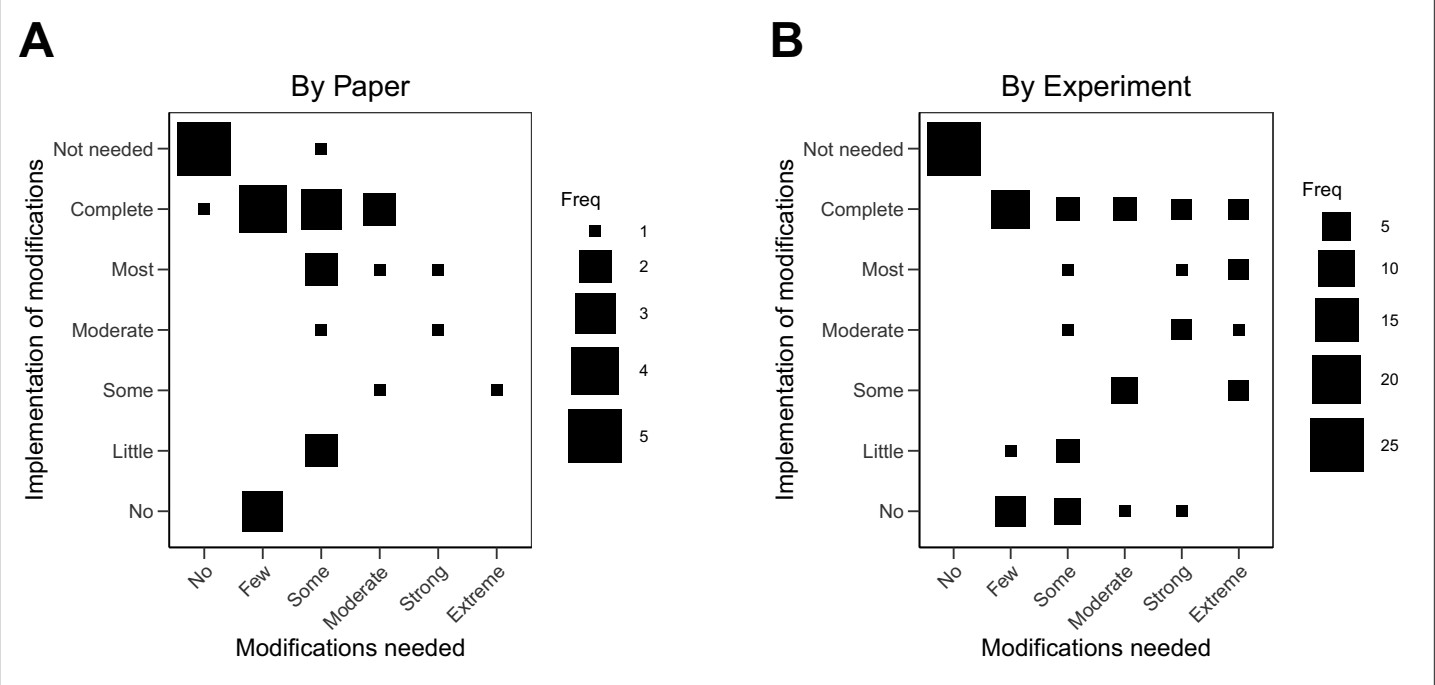

**Figure 3.** Relationship between extent of modifications needed and implementation of modifications. Fluctuation plots showing the coded ratings for extent of modifications needed in order to conduct the replication experiments, and the extent to which the replication authors were able to implement these modifications for experiments that were conducted. The size of the square shows the number (Freq) of papers/experiments for each combination. (**A**) To characterize papers (N = 29), coded ratings were averaged across the experiments conducted for each paper. The average number of experiments conducted per paper was 2.6 (SD = 1.3; range = 1–6), and the Spearman rank-order correlation between extent of modifications needed and implementation was –0.01 (95% CI [–0.42, 0.40]). (**B**) For the experiments that were started (N = 76), the Spearman rank-order correlation was 0.01 (95% CI [–0.27, 0.28]).

falsify the original claim because the replication does not test it. But, depending on the reasons for failure in implementation, it could raise doubt about the reliability of the claim if it seems that the original methodology could not have produced the reported outcomes either. For example, if the replication study suggests that the model system does not behave as reported in the original study, it could indicate a flaw in the original study or an unknown feature that is necessary to obtain the original behavior.

### The challenges encountered during peer review of the Replication Studies

In total, we completed 50 experiments from 23 of the original papers (*Table 1*). This means that no experiments were completed for six of the original papers for which Registered Reports were published. For 18 of the original papers we were able to complete all experiments described in the Registered Report, so for each of these we prepared and submitted a Replication Study that reported the results of the completed experiments. For five of the original papers we were only able to complete some of the experiments

described in the Registered Report: in these cases the results of the completed experiments were reported in an aggregate paper (*Errington et al., 2021a*).

In the Registered Report/Replication Study model (https://cos.io/rr/), peer review of the Replication Study is supposed to be independent of outcome to mitigate publication bias, suppression of negative results, and results-contingent motivated reasoning interfering with publication (*Nosek and Lakens, 2014*; *Chambers, 2019*). Reviewers examine whether the authors completed the experiments as proposed, appropriately interpreted the outcomes, and met any outcome-independent quality control criteria that were defined during the review of the Registered Report. Usually the review process played out according to these ideals, occasionally it did not. This is understandable, partly because the Registered Report model is new for many reviewers, and partly because when observed outcomes differ from expectations it provokes immediate reasoning and rationalizing about why it occurred (*Nosek and Errington, 2020a*). Indeed, such interrogation of unexpected outcomes is

## Box 6. Case study: Peer review of protocols prior to conducting the experiments and after the results are known.

The replication protocol (*Vanden Heuvel et al., 2016*) for testing the sensitivity of Ewing's sarcoma cell lines to PARP inhibitors was based on the original paper (*Garnett et al., 2012*), like all replication protocols. The original authors provided additional feedback to ensure a good-faith replication protocol. Peer review of the protocols further increased the rigor and accuracy of the experimental designs, such as including additional measurements of proliferation to ensure all cell lines were replicating at the time of drug treatment and specifying the minimal number of colonies in the control condition before stopping the experiment. Peer review of the protocols also acknowledged the challenge we faced of not having access to all of the exact same cell lines as the original study and did not raise any concerns when cell lines of the same disease/status were proposed (e.g., different Ewing' sarcoma cell lines). After the experiments were conducted, the results were submitted for peer review, and the reviewer comments were largely focused on trying to reconcile the differences between the results of the original study and the results of the replication (*Vanden Heuvel et al., 2018*). A lack of concern about inexact comparability of cell lines in the reviews before the results were known was replaced with highlighted concern that this difference accounted for the lack of statistically significant results in the replication after the results were known. Similarly, after the fact, reviewers raised concerns about the timing of an experiment as potentially not allowing for the effect to be measurable due to the need for cells to be in a proliferative state despite the fact that the design was identical between the replication and original experiments. Some speculations, such as the use of different sarcoma cell lines and the level of knockdown efficiency, are possible explanations for the different results in the replication experiments, but they require follow-up tests to assess whether they actually account for the observed differences. We included these possibilities when discussing the results in the Replication Study, but disagreed with a request from the reviewers that the speculations justified labeling the replication result as "inconclusive".

productive for hypothesis generation and exploration of what could be studied next.

A presumed virtue of Registered Reports is that it incorporates preregistration (*Camerer et al., 2018*) to very clearly separate hypothesis testing (confirmatory) and hypothesis generating (exploratory) modes of analysis. Another virtue is that expert feedback is incorporated during design to improve the quality of the experiments (*Soderberg et al., 2021*). During peer review of the Registered Reports the reviewers ensure that the proposed experiments are appropriately designed and fair tests of the original findings. That precommitment, by both replication authors and reviewers, is a mechanism to ensure that all results are taken seriously whether they confirm or disconfirm the original finding (*Nosek and Errington, 2020b*).

During peer review of the Replication Study, the authors and reviewers observe the outcomes and wrestle with what they mean. But, because they made precommitments to the experiments being diagnostic tests of the original finding, new ideas that occur following observation of the outcomes are clearly designated as hypothesis generating ideas for what should be studied next. For example, when an outcome is inconsistent with the original finding, it is common for reviewers to return and re-evaluate methodology (see Case study in *Box 6*). Features or differences from the original experiments that seemed immaterial a priori become potentially important *post facto*. The risk, of course, is that the *post facto* reasoning is just rationalization to maintain belief in an original finding that should now be questioned (*Kerr, 1998*; *Kunda, 1990*). Registered Reports mitigates that risk with the precommitment to publishing regardless of outcomes, and then speculations for the causes of different results from the original experiments can be actively and independently tested to assess their veracity.

This mostly occurred as intended in this project. Of the 18 Replication Studies submitted

to *eLife*, 17 were accepted and one was rejected. The rejected Replication Study was posted as a preprint (*Pelech et al., 2021*). *eLife* makes reviewer comments and author responses to reviews public with the published papers. Links to all published papers and reviewer comments are in *Table 1*. With rejection of one completed Replication Study, the Registered Reports model was mostly effective at eliminating publication bias against negative results (*Allen and Mehler, 2019*; *Scheel et al., 2020*). With peer review in advance, the Registered Reports model was effective at fostering precommitments among authors and reviewers to the replication experimental designs (*Nosek and Errington, 2020b*). And, as evidenced by the diversity of reactions in the open reviews and commentaries on the final Replication Studies, the Registered Reports model did not eliminate divergence and disagreement among researchers about the meaning and implications of the replication

findings. As long as all outcomes are reported, such divergence after the fact may be productive for stimulating critical inquiry and generating hypotheses even when it is indicative of intransigence or motivated reasoning to preserve prior claims.

### The duration of the different phases in the project

On average the gap between paper selection and the submission of a Registered Report was 30 weeks (mean), and the gap between submission and acceptance for publication was 19 weeks (*Figure 4*). It then took an average of 12 weeks to prepare experiments for data collection. The gap between the start of experimental work and final data delivery was 90 weeks, and another 24 weeks were needed to analyse the data and write the Replication Study. The gap between submission of the Replication Study and

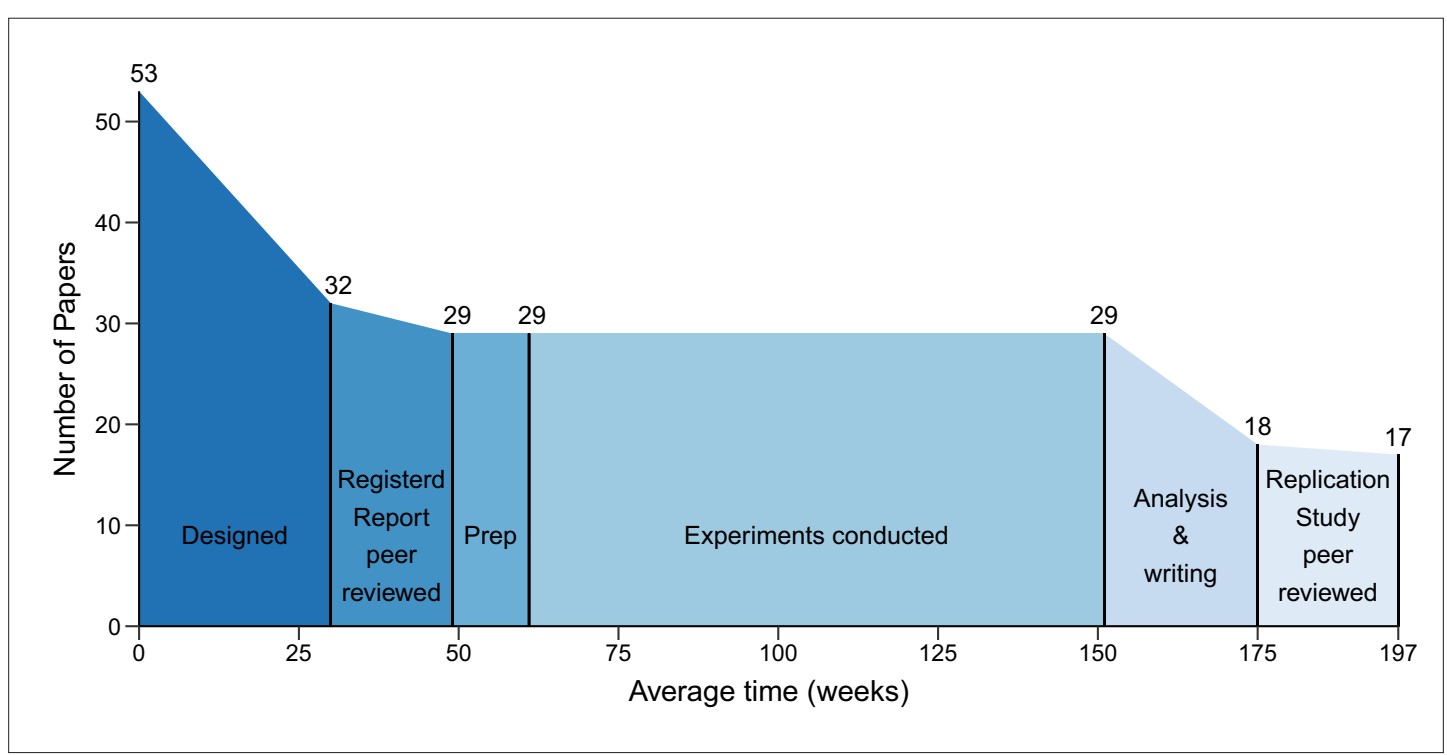

**Figure 4.** The different phases of the replication process. Graph showing the number of papers entering each of the six phases of the replication process, and the mean duration of each phase in weeks. 53 papers entered the design phase, which started with the selection of papers for replication and ended with submission of a Registered Report (mean = 30 weeks; median = 31; IQR = 21–37). 32 papers entered the protocol peer reviewed phase, which ended with the acceptance of a Registered Report (mean = 19 weeks; median = 18; IQR = 15–24). 29 papers entered the preparation phase (Prep), which ended when experimental work began (mean = 12 weeks; median = 3; IQR = 0–11). The mean for the prep phase was much higher than the median (and outside the IQR) because this phase took less than a week for many studies, but much longer for a small number of studies. The same 29 papers entered the conducted phase, which ended when the final experimental data were delivered (mean = 90 weeks; median = 88; IQR = 44–127), and the analysis and writing phase started, which ended with the submission of a Replication Study (mean = 24 weeks; median = 23; IQR = 7–32). 18 papers entered the results peer review phase, which ended with the acceptance of a Replication Study (mean = 22 weeks; median = 18; IQR = 15–26). In the end, 17 Replication Studies were accepted for publication. The entire process had a mean length of 197 weeks and a median length of 181 weeks (IQR = 102–257).

acceptance for publication was 22 weeks. On average the process took 197 weeks.

All the experimental details (e.g., additional protocol details, data, analysis files) are openly available at https://osf.io/collections/rpcb/ (see *Table 1* for links to individual studies), or domain specific repositories (e.g., https://www.metabolomicsworkbench.org); physical materials (e.g., plasmids) were made openly available where possible (e.g., https://www.addgene.org).

## Discussion

Much of the concern about replicability in science is whether reported findings are credible (*Begley and Ellis, 2012*; *Camerer et al., 2016*; *Camerer et al., 2018*; *Errington et al., 2021b*; *Open Science Collaboration, 2015*; *Prinz et al., 2011*). Our experience conducting this project identifies a much more basic and fundamental concern about replication – it is hard *to assess* whether reported findings are credible. We attempted to replicate 193 experiments from 53 papers, but we experienced reproducibility challenges at every phase of the research lifecycle. Many original papers failed to report key descriptive and inferential statistics including 27% of experiments just presenting representative images and 21% of experiments reporting inferential test outcomes not reporting which test was conducted. Raw data was publicly accessible for just 2% of experiments to reproduce the findings, compute effect sizes, and conduct power analyses. After requesting original data from authors, we acquired raw data for 16%, summary data for 14%, and nothing for 68% of experiments. None of the 193 experiments was described completely enough to design a replication protocol without requesting clarifying details from the original authors.

Authors were bimodal in their helpfulness in sharing data and materials and providing feedback, 32% were *not at all helpful/no response* and 26% were *extremely helpful*. Implementation of peer-reviewed and preregistered protocols often led to unexpected challenges such as model systems behaving differently than originally reported, requiring modifications to protocols. Just 33% of experiments required no modifications. Of those needing modifications, 41% were implemented completely. Cumulatively, these process challenges for assessing replicability slowed the project and increased costs (see Appendix 1). After an extended data collection period, we completed replications of 50 experiments from 23 papers.

Original papers do not include enough information about the methodology and results. Original data and materials are not archived and accessible in repositories. Original authors are variably willing or able to clarify information gaps and share data, materials, and reagents to facilitate assessment of original findings. These challenges slowed progress, inflated costs, and made it harder to design and conduct replication studies. None of this is a direct indication of whether any particular original finding is credible, but all of it is a challenge for credibility of research in general (*Begley and Ioannidis, 2015*; *Ioannidis et al., 2014*). Credibility of scientific claims is rooted in their independent verifiability (*Nosek and Errington, 2020a*; *Putnam, 1975*; *Schmidt, 2009*). Pervasive impediments to verification mean that research is not living up to the "show me" ethos of science and is functionally operating as a "trust me" enterprise.

Practical barriers to the assessment of replicability compounds the credibility risk that is already present with a research culture that prizes innovation at the expense of verification (*Martin, 1992*; *Sovacool, 2008*). Publication is achieved, grants are given, and careers are made on the production of positive results not negative results, tidy evidence and explanation not uncertainty and exceptions, and novel findings not replications or incremental extensions of prior work (*Giner-Sorolla, 2012*; *Mahoney, 1977*; *Nosek et al., 2012*). These incentives encourage publication bias against negative results and selective reporting to indicate stronger, cleaner findings than the reality of the evidence – and the behaviors that produce these outcomes could occur without intention or control via motivated reasoning (*Hart et al., 2009*; *Kunda, 1990*), confirmation bias (*Nickerson, 1998*), and hindsight bias (*Christensen-Szalanski and Willham, 1991*; *Fischhoff and Beyth, 1975*). Lack of documentation and transparency of the research process makes it difficult to identify these behaviors. And, even if researchers are motivated to conduct independent verification, not only are there disincentives to spend resources on reproduction and replication and cultural resistance to the practice, there are also mundane practical barriers to doing so because of lack of documentation, transparency, and sharing. In short, we have created a research culture in which assessing replicability and reproducibility is unrewarded, unnecessarily difficult, and potentially career damaging.

If the published literature were highly credible, and if false starts were efficiently weeded out of

the literature, then the lack of reward and feasibility for verification and replication efforts might not be a cause for concern. However, the present evidence suggests that we should be concerned. As reported in *Errington et al., 2021b*, replication efforts frequently produced evidence that was weaker or inconsistent with original studies. These results corroborate similar efforts by pharmaceutical companies to replicate findings in cancer biology (*Begley and Ellis, 2012*; *Prinz et al., 2011*), efforts by a non-profit biotech to replicate findings of potential drugs in a mouse model of amyotrophic lateral sclerosis (*Perrin, 2014*), and systematic replication efforts in other disciplines (*Camerer et al., 2016*; *Camerer et al., 2018*; *Cova et al., 2018*; *Ebersole et al., 2016*; *Ebersole et al., 2019*; *Klein et al., 2014*; *Klein et al., 2018*; *Open Science Collaboration, 2015*; *Steward et al., 2012*). Moreover, the evidence for self-corrective processes in the scientific literature is underwhelming: extremely few replication studies are published (*Makel et al., 2012*; *Makel and Plucker, 2014*); preclinical findings are often advanced to clinical trials before they have been verified and replicated by other laboratories (*Chalmers et al., 2014*; *Drucker, 2016*; *Ramirez et al., 2017*); and many papers continue to be cited even after they have been retracted (*Budd et al., 1999*; *Lu et al., 2013*; *Madlock-Brown and Eichmann, 2015*; *Pfeifer and Snodgrass, 1990*). If replicability is low and the self-correction processes in science are not efficiently separating the credible from the not credible, then the culture of modern research is creating unnecessary friction in the pace of discovery.

Fundamentally, the problem with practical barriers to assessing replicability and reproducibility is that it increases uncertainty in the credibility of scientific claims. Are we building on solid foundations? Do we know what we think we know? Assessing replicability and reproducibility are important mechanisms for identifying whether findings are credible, for clarifying boundary conditions on circumscribed findings, and for generalizing findings to untested circumstances (*Nosek and Errington, 2020a*). There are open questions about the appropriate distribution of resource investment between innovation and verification efforts. Here, for example, though costs increased because of the unexpected impediments, the final cost per experiment of approximately $53,000 might be seen as comparatively modest compared to the losses incurred by follow-on research for findings that are unreplicable or much more limited than

initially believed. DARPA's Friend or Foe program might be a case in point in which a portion of the program budget is invested in independent verification and validation (*Raphael et al., 2020*). In any case, an efficient science would not impose unnecessary practical barriers to verification just as it should not impose unnecessary practical barriers to innovation. We can do better. Fortunately, there are mechanisms that could greatly enhance the ability to assess whether reported findings are credible, and reduce the barriers to verification efforts more generally. Moreover, some mechanisms are in practice already demonstrating their feasibility for broad implementation.

## Improving documentation and reporting

Reading the paper and supplementary materials was sufficient to design the replication study for none of the 193 experiments. Lack of interest or attention to methods, space constraints, and an absence of standards may all contribute to weaknesses in documentation of how the research was done. Better reporting will improve research efficiency by helping authors and peer reviewers identify errors or other potential limitations. Better reporting will also improve research efficiency by helping readers who wish to replicate or extend the research to develop accurate experimental designs. Our sample of papers came from articles published between 2010 and 2012. Since then, some publishers have taken steps to improve reporting and standards have emerged to promote consistency, clarity, and accuracy (*Marcus, 2016*; *Nature, 2013*), and a coalition of publishers and other stakeholders are promoting minimum reporting standards for life science (*Macleod et al., 2021*). Also, increasing frequency of citation of data, materials, reagents, and antibodies highlights improving reporting standards (*Han et al., 2017*; *Macleod, 2017*).

There is still a long way to go before strong reporting is normative (*Baker et al., 2014*; *Gulin et al., 2015*), but the efforts to establish reporting standards and requirements has positioned the community for significant improvement in making it possible to understand how the research was conducted (*Glasziou et al., 2014*; *Macleod et al., 2014*). A potential negative consequence of improving documentation and reporting is additional burden on researchers without compensatory benefits for actually improving research. Regardless of their benefits, implementations of reporting standards should make them easy and efficient to adopt and attentive to diminishing returns. The sweet spot

of reporting standards is to provide sufficient structure, specificity, and support to make the research process transparent and simultaneously to avoid turning a good practice into just another bureaucratic burden.

### Improving data, code, and materials transparency and sharing

For many of the experiments we examined, we could not determine key details of the original results from the paper such as sample size, effect size, or variability. Data and code were almost never available in public repositories, and requests for sharing the original data mostly failed. It is not possible to assess reproducibility or robustness if data are not available. And, policies that data are to be made available "upon request" are recognized as ineffective (*McNutt et al., 2016*). One obvious reason is that such requests come long after the original researchers have moved on from the project, making the data difficult, impossible, or time-consuming to recover. Hundreds of journals have strengthened their policies to promote data and code sharing, and the rates of sharing are improving, if slowly (*Camerer et al., 2018*; *Serghiou et al., 2021*; *Stodden et al., 2013*; see journal transparency policies at https://topfactor.org). The infrastructure for sharing and archiving data and code has blossomed with domain-specific repositories for a wide variety of data types such as GenBank, Protein DataBank, and Cancer Imaging Archive, and emergence of metadata standards more generally (*Wilkinson et al., 2016*). Generalist repositories such as OSF, Zenodo, and Figshare offer archiving solutions for digital data of almost any kind.

Repositories are likewise available for sharing digital materials such as protocols, additional images, IACUC or IRB documentation, or any other content. For example, the OSF projects for these replication efforts include cell line authentication (e.g., STR and mycoplasma testing), plasmid verification (e.g., sequencing files), maintenance records (e.g., cell culture, animal husbandry), and all raw images (e.g., Western blot, immunohistochemistry, bioluminescence images) for relevant experiments alongside the data and code. Another challenge to address to improve research efficiency is burdens and delays for sharing physical materials such as cells, plasmids, animals, and antibodies. Repositories are available for sharing physical materials (e.g., https://www.addgene.org, https://www.mmrrc.org, https://www.atcc.org) and relieves

scientists of having to maintain and distribute to other researchers minimizing costs associated when researchers have to make them again (see Case study in *Box 3*; *Lloyd et al., 2015*). When not available in a repository, we experienced a variety of unnecessary barriers and delays navigating material transfer agreements with institutions because of lack of interest, infrastructure, or policy for facilitating material sharing. There is an opportunity for substantial improvement not only for replications but also for novel research that builds upon published research by having better funding and legal structures for sharing materials. For example, the initiation of replication experiments at replication labs was significantly accelerated by the existence of standard master services agreements already in place with all replicating labs via the Science Exchange marketplace.

Potential negative consequences of improved sharing can occur if the scholarly reward systems fail to catch up. At present, some researchers see risk and little reward for sharing because of lack of credit for doing so. Evidence suggests that there is more benefit than cost (*McKiernan et al., 2016*), but altering reward systems toward treating data, materials, and code as citable scholarly contributions will ease the perceived risks.

### Improving preregistration of experiments and analysis plans

Two key factors undermining the credibility and replicability of research are publication bias and questionable research practices like *p*-hacking. With publication bias, negative findings are much less likely to be reported than positive findings (*Greenwald, 1975*; *Rosenthal, 1979*). With questionable research practices, discretion in data analysis and selective reporting of outcomes can lead to intentional or unintentional manufacturing and reporting of positive outcomes that are more favorable for publication (*Casadevall and Fang, 2012*; *Gelman and Loken, 2013*; *Ioannidis, 2005*; *John et al., 2012*; *Kaplan and Irvin, 2015*; *van der Naald et al., 2020*; *Simmons et al., 2011*). These lead to a biased literature with exaggerated claims and incredible evidence (*Begley and Ellis, 2012*; *Open Science Collaboration, 2015*; *Prinz et al., 2011*; *Smaldino and McElreath, 2016*).

One solution to these challenges is preregistration (*Nosek et al., 2019*; *Nosek et al., 2018*; *Wagenmakers et al., 2012*). Preregistration of experiments mitigates publication bias by making all research discoverable whether or not it is

ultimately published. Preregistration of analysis plans solves selective reporting by making clear what analyses were planned a priori and what was determined and conducted after the fact. Planned and unplanned analyses both contribute to advancement of knowledge, the latter often being the source of unexpected discoveries. But, unplanned analyses that occur after observing the data are usually more tentative and uncertain. Preregistration helps increase visibility of that uncertainty and reduce the likelihood of inadvertently mistaking an uncertain exploratory result as a confirmatory test of an existing hypothesis. For areas of research that have been investigating replicability, such as psychology and economics, preregistration has gained rapid adoption (*Christensen et al., 2019*; *Nosek and Lindsay, 2018*).

Preregistration is still relatively rare in basic and preclinical research in the life sciences, but the potential for improving replicability and research efficiency is pronounced. In life science experiments involving animals, there are significant ethical implications for not publishing negative results derived from these experiments. Recent studies suggest the data from only 26% of animals used in life science experiments are ever published (*van der Naald et al., 2020*). One could argue that ensuring outcome reporting of all animal experiments is an ethical issue, and IACUC's could incentivize or require preregistration as a compliance mechanism. A recent NIH committee report focusing on improving research rigor and reproducibility recommended piloting preregistration in animal research to test its effectiveness (*Wold et al., 2021*).

Like improving reporting standards, a potential risk of preregistration is creating bureaucratic burden that does not exceed the benefits of instituting the process. Technology supporting preregistration can minimize that burden with efficient workflows that researchers perceive as supporting effective research planning rather than imposing reporting burdens. Also, misperceptions that preregistration discourages exploratory or discovery oriented research could interfere with effective adoption and application. As such, education and training are essential components of effective adoption.

### Improving rigor, reporting, and incentives with Registered Reports

All replication studies were peer reviewed at *eLife* prior to conducting the research, a publishing model called Registered Reports. With Registered Reports, expert critique improves experimental designs before they are conducted rather than just pointing out the errors and problems after the work is completed. Preregistration is built into the process eliminating publication bias and providing a clear distinction between planned analyses and exploratory discoveries. Publication decisions are made based on the importance of the research question and the quality of the methodology proposed to test the question, not whether the observed outcomes are exciting or as expected. Incentives for exciting findings, regardless of credibility, are removed. Researchers are instead incentivized to ask important questions and design creative and compelling tests of those questions (*Chambers, 2019*).

As of late 2021, more than 300 journals have adopted Registered Reports as a submission option, mostly in the social-behavioral sciences and neuroscience. Evidence to date suggests that Registered Reports are effective at eliminating publication bias. In a sample of 71 Registered Reports and 152 comparison articles from the same outlets published around the same time, 56% of primary outcomes were negative results for Registered Reports and 4% for comparison articles (*Scheel et al., 2020*). Moreover, despite the increase of supposedly "boring" negative results, a sample of Registered Reports received similar or greater altmetric attention and citation impact as comparison articles (*Hummer et al., 2017*). An observational study also found evidence that Registered Reports outperform comparison articles on all 19 outcome measures from slightly on measures of novelty and creativity to strongly on measures of quality and rigor (*Soderberg et al., 2021*).

Some funders and journals are conducting partnerships via Registered Reports in which a single peer review process results in in-principle acceptance of the paper and funding to conduct the experiments such as programs sponsored by the Children's Tumor Foundation (https://www.ctf.org/research/drug-discovery-initiative-registered-reports-ddirr) and The Flu Lab (https://cos.io/flulab/). This offers a compelling incentive alignment for researchers and opportunity for journals to receive and publish high-quality, funded projects and funders to maximize their return on investment by ensuring that funded studies don't wind up in the file drawer. Like preregistration, a potential unintended negative consequence of Registered Reports is if the model shifts the culture away from valuing exploratory and discovery-oriented research. Ideally, both practices facilitate clarity of when

research is testing versus generating hypotheses without fostering the perception that research progress can occur with one and without the other.

### Improving incentives for replication

With a research culture that prizes innovation and novelty, verification and replication gets pushed aside. Innovation without verification creates a fragile and fragmented evidence base that may slow the pace of knowledge accumulation (*Chalmers et al., 2014*). Replication is essential for advancing theory because it provides an opportunity to confront and refine current understanding (*Nosek and Errington, 2020a*). Investigations of the prevalence of replication studies in the published literature yield extremely low estimates in different disciplines (*Makel and Plucker, 2014*; *Makel et al., 2012*; *Pridemore et al., 2018*; *Valentine et al., 2011*). There is no known systematic investigation of the prevalence of replication studies in cancer biology, but like other fields – with a strong emphasis on innovation and novelty in cancer biology – there is little encouragement by journals or funders for proposing, conducting, or reporting replications. Without reward systems for replication research, it is unlikely that the near exclusive emphasis on innovation will be reduced.

Simultaneously, it is not clear that a dramatic shift in the proportion of studies for replication is needed. Only a small portion of the research literature has a substantial impact on the direction and investment in research. By focusing replication resources on the research that is having significant impact and spurring new investment, even a small infusion of funding, journal space, and institutional reward for replications could have a dramatic effect on improving clarity about credibility and replicability – emboldening investments on productive paths and saving resources from dead ends. That is not to suggest that replications provide definitive evidence to confirm or disconfirm original findings. Rather, successful replications promote confidence that new findings are reliable and justify further investigation into their validity and applicability, and unsuccessful replication prompt questions to look closer at the phenomenon to determine whether the failure is due to a false positive in the original research, a flaw in the replication, or previously unidentified conditions that influence whether the phenomenon is observed (*Errington et al., 2021b*; *Nosek and Errington, 2020b*; *Nosek and Errington, 2020a*).

There is some movement toward valuing replication research more explicitly with some journals providing explicit statements in their policies about publishing replications and funders like DARPA in the US explicitly investing in independent verification and validation as part of ongoing programs pursuing research innovations (*Raphael et al., 2020*). Also, funders are occasionally launching programs to support replication studies such as the NWO in the Netherlands (*Baker, 2016*) and the NSF in the US (*Cook, 2016*). A potential risk of increased rewards for replication is if expectations for conducting or achieving replicability become so high that they discourage risk-taking and pursuit of highly resource intensive investigations. For example, in early phases of research, low replicability is not surprising because researchers are often pursuing ideas that have low prior odds of success. The optimal mixture of investment in innovation versus verification research is unknown.

Collectively, these improvements to transparency and sharing are captured by the Transparency and Openness Promotion Guidelines (TOP; https://cos.io/top), a policy framework for publishers, funders, and institutions to set standards for transparency of the research process and outputs by their authors, grantees, or staff (*Camerer et al., 2018*). As of 2020, more than 1,000 journals have implemented TOP compliant policies for one or more of the categories of improvements. TOP Factor (https://topfactor.org) rates journal policies on promoting transparency, openness, and reproducibility, and the web interface makes it easy to compare across journals. Pervasive adoption of open behaviors and policies by all stakeholders would help shift norms and set higher standards for transparency, sharing, and reproducibility of research.

Simultaneously, an active metascience research community that evaluates the impact of these new behaviors and policies will help identify unintended negative consequences, improve their implementation, and optimize their adoption for facilitating research progress. Stakeholders in the cancer biology community including researchers, funders, societies, and institutional representatives could facilitate and support research investigations of the scientific process so that decisions about adopting these behaviors at scale can be evidence-based and clearly represent both the costs and benefits.

## Conclusion

We experienced substantial challenges when designing protocols to replicate experiments from published papers because the papers often did not contain the information required for such replications (such as raw data and identifiers for reagents and research materials). There is substantial opportunity to improve the seemingly mundane but critical behaviors of documentation, transparency, and open sharing of protocols, data, and research materials, if the scientific community is to improve the reproducibility, replicability, and reuse of research. Initiatives to improve the reporting of methods and results – including preregistration of experiments, and the reporting of both negative and positive results – have started to make headway in the life sciences, and some are becoming mainstream in neighboring disciplines. Collectively, these initiatives offer substantial opportunities to improve the replicability and credibility of research and, ultimately, to advance scientific knowledge in a way that is both efficient and reliable.

## Materials and methods

### *Paper and experiment selection strategy*

50 papers published in 2010, 2011 or 2012 were selected as described in *Errington et al., 2014*. After the project started one paper was replaced because it contained sequencing and proteomic experiments and should not have been selected in the first place. During the course of the project, after contacting the original authors, we determined that it would not be feasible to conduct replications for three papers, so these papers were replaced.

Experiments for replication were identified as described in *Errington et al., 2014*. Corresponding authors were contacted and shared the drafted replication protocols based on information from the original papers. Specific questions were highlighted including requests for original data, key materials that were identified, and protocol clarifications. We also asked for any additional information that could improve the quality of the replication attempt. Following initial author feedback, we shared replication protocols with research providers from the Science Exchange marketplace, which consists of a database of searchable scientific service providers that have been qualified and contracted under a standard already negotiated master services agreement. On average it took 6 days from placing requests to receiving a quote from replicating labs (median

= 2 days; IQR = 1–8). In total 48 providers participated in the project (22 academic shared resource facilities and 26 contract research organizations [CROs]) by reviewing and contributing to replication protocols, including describing deviations from the original study (e.g., different instrumentation), and conducting the replication experiments themselves. Experimental designs and protocols were iterated based on comments and suggestions from original authors, when possible, and the replicating researchers. Experiments were then submitted as a Registered Report to *eLife* where it underwent peer review and if approved began experimentation. In all, 193 experiments were included in the project: 188 experiments were identified at the start of the project; three were added during peer review of the Registered Reports, and two were added following the exchange of comments and suggestions with original authors. At the same time, 83 experiments were dropped following exchanges with original authors. Of the 110 experiments that continued, 97 were included in Registered Reports that we submitted to *eLife*. The 29 Registered Reports that were accepted for publication included 87 experiments.

### *Coding*

Papers were coded for metadata and whether corresponding authors responded to any email requests. Experiments were coded on a number of variables from the papers, requests and input from the original authors, and information about the replication attempt. Experiments were linked to specific figures and tables in the original papers. Variables were coded as described in the data dictionary and figures, figure legends, main text, methods, and supplementary figures/tables were searched for the information. Variables about requests and input from the original authors were coded based on the protocol documents shared with original authors for input and the responses received and were either objective or subjective. Information about the replication attempts were coded based on objective features or our subjective experience of the process. For subjective variables coded responses were given according to a Likert scale with examples given in the main text to provide illustrations of the subjective coding. Data dictionaries describing all of the variables are available at https://osf.io/e5nvr/.

### *Statistical analysis and reporting*

Descriptive and exploratory statistics were used to analyze coded variables in R software

(RRID:SCR_001905), version 4.0.3 (*R Development Core Team, 2021*). *Figures 2–4* were generated using the ggplot2 (version 3.3.3) package. Exploratory analysis (Spearman rank-order correlation) was conducted after data were checked to ensure assumptions were met.

## Note

All *eLife* content related to the Reproducibility Project: Cancer Biology is available at: https://elifesciences.org/collections/9b1e83d1/reproducibility-project-cancer-biology.

All underlying data, code, and digital materials for the project is available at: https://osf.io/collections/rpcb/.

## Acknowledgements

This work was supported by a grant from Arnold Ventures (formerly known as the Laura and John Arnold Foundation), provided to the Center for Open Science in collaboration with Science Exchange. We thank Anne Chestnut for assistance creating *Figure 1*. We thank Fraser Tan, Joelle Lomax, Rachel Tsui, and Stephen Williams for helping in coordination efforts during the course of the project. We thank all Science Exchange providers who provided their services, and all employees at Science Exchange and the Center for Open Science who contributed to administrative and platform development efforts that enabled this project to occur.

**Timothy M Errington** is at the Center for Open Science, Charlottesville, United States
tim@cos.io
(iD) http://orcid.org/0000-0002-4959-5143
**Alexandria Denis** is at the Center for Open Science, Charlottesville, United States
**Nicole Perfito** is at Science Exchange, Palo Alto, United States
(iD) http://orcid.org/0000-0001-9546-215X
**Elizabeth Iorns** is at Science Exchange, Palo Alto, United States
(iD) http://orcid.org/0000-0002-5515-1258
**Brian A Nosek** is at the Center for Open Science and the University of Virginia, Charlottesville, United States
(iD) http://orcid.org/0000-0001-6797-5476

*Author contributions:* Timothy M Errington, Conceptualization, Data curation, Formal analysis, Investigation, Methodology, Project administration, Supervision, Validation, Visualization, Writing – original draft, Writing – review and editing; Alexandria Denis, Data curation, Investigation, Writing – review and editing; Nicole Perfito, Conceptualization, Investigation, Methodology, Project administration, Writing – review and editing; Elizabeth Iorns, Conceptualization, Data curation, Funding acquisition, Investigation, Methodology, Project administration, Supervision, Validation, Writing – review and editing; Brian A Nosek, Conceptualization, Funding acquisition, Methodology, Supervision, Writing – original draft, Writing – review and editing

*Competing interests:* Timothy M Errington: Employed by the nonprofit Center for Open Science that has a mission to increase openness, integrity, and reproducibility of research. Alexandria Denis: Employed by the nonprofit Center for Open Science that has a mission to increase openness, integrity, and reproducibility of research. Nicole Perfito: Employed by and hold shares in Science Exchange Inc. Elizabeth Iorns: Employed by and hold shares in Science Exchange Inc. Brian A Nosek: Employed by the nonprofit Center for Open Science that has a mission to increase openness, integrity, and reproducibility of research.

## Funding

| Funder | Grant reference number | Author |
|---|---|---|
| Arnold Ventures | | Timothy M Errington Alexandria Denis Nicole Perfito Elizabeth Iorns Brian A Nosek |

The funders had no role in study design, data collection and interpretation, or the decision to submit the work for publication.

## Decision letter and Author response

Decision letter https://doi.org/10.7554/eLife.10.7554/eLife.67995.sa1
Author response https://doi.org/10.7554/eLife.10.7554/eLife.67995.sa2

## Additional files

### Supplementary files

• Transparent reporting form

### Data availability

All experimental details (e.g., additional protocol details, data, analysis files) of the individual replications and data, code, and materials for the overall project are openly available at https://osf.io/collections/rpcb/; see Table 1 of the present article for links to individual studies. Master data files, containing the aggregate coded variables, are available for exploratory analysis at https://osf.io/e5nvr/.

The following dataset was generated:

| Author(s) | Year | Dataset URL | Database and Identifier |
|---|---|---|---|
| Errington TM, Denis A | 2021 | https://osf.io/e5nvr/ | Open Science Framework, e5nvr |

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

## Appendix 1

### Costs associated with evaluating reproducibility

In conducting research, there is a constant interplay of evaluating available resources for making decisions about research investments. This is a combination of time, cost, and accessibility of materials and reagents to do the research. In typical work, that decision matrix also includes questions about potential viability of the research direction, potential impact of success, and confidence in the current evidence (shall we replicate or proceed with the evidence we have?). These latter items were not complicating factors for our project, but the former ones were substantial as we had many papers, many experiments, lab sourcing, design feedback, cost projections, time projections, and coordination decisions to resolve. One of the most concrete ways to express that challenge is the evolving cost estimates for conducting the research over the course of the project.

At the start of the project we budgeted $25,000 per paper, and updated this figure as the project progressed. By the time peer review of the Registered Reports began, the estimated cost per paper had increased to $35,750 (median = $33,822; IQR = $26,448–$44,260). At the onset of data collection, the average estimate was $42,017 (median = $39,843; IQR = $28,069–$55,750). And, the actual average cost on completion was $52,574 per replication study (median = $53,089; IQR = $33,994–$61,496). In total, $1,524,640 was spent on replication experiments. Not included in these costs are project administration costs, particularly personnel costs, accrued as the project took longer to complete than originally estimated because of the unexpected challenges of getting feedback, obtaining materials, carrying out the experiments, internal delays in project management, and the common delays in peer review. Additionally, not counted in these costs are donated reagents from scientific suppliers and replicating labs that provided discounted costs to support the project.

Delays and increasing costs were practical challenges for investigating reproducibility. If data and materials were readily available, the time and cost of designing experiments would have been lower. If original papers were more comprehensive in reporting methodology, the time and cost of designing protocols would have been lower. If providing feedback on replication designs were normative, the time and cost of confirming protocols would have been lower. Improving sharing, documentation, and feedback are fixable with changes to norms and policies.

Some delays and costs were due to experimental systems (e.g., tumor growth in animals) not behaving as they had in the original study. Whether or not it is possible to make experimental systems more consistent in their behavior depends on whether these inconsistencies are an inherent feature of working with complex biological systems that cannot be avoided, or if they are due to weaknesses in methodology that could be addressed.

Delays and increasing costs also had other consequences: for example, experiments were staggered over time, so as some experiments were completed, we were able to update time and cost estimates for experiments not yet started. Decisions to end individual experiments were influenced partly by challenges unique to that experiment, and partly by factors related to time and cost estimates across the project as a whole. A decision to end an experiment does not necessarily mean that the original finding is unreplicable. It is possible that devoting more project resources to any one finding would have ultimately resolved the challenges and replicated the result successfully. Likewise, a decision to end an experiment does not validate the original finding. It is possible that the practical challenges are indicators of deeper issues with the original findings.

