## [Decision Letter]

**Decision letter after peer review:**

Thank you for submitting your article "Challenges for Assessing Reproducibility and Replicability Across the Research Lifecycle in Preclinical Cancer Biology" to *eLife* for consideration as a Feature Article. Your article has been reviewed by three peer reviewers, including the *eLife* Features Editor Peter Rodgers and the evaluation has been overseen by Eduardo Franco as the Senior Editor.

The reviewers and editors have discussed the reviews and we have drafted this decision letter to help you prepare a revised submission.

Summary:

This is a well written paper that clearly expresses the various issues concerning interactions with scientists to acquire reagents etc. to test data reproducibility for "high impact" papers published during 2010-2012.

For the purpose of this review, I refer to the authors of the series as the "replicators" to distinguish them from the "authors" of the 2010-2012 publications. The current manuscript summarizes the heterogeneity of the outcomes of their communications with the authors of the papers whose experiments they sought to replicate. Many authors do not appear to have been willing to provide additional data when requested by the replicators. In this regard, Figure 3 presents an interesting analysis that suggests a limited correlation between the amount of information requested by the replicators and the degree of cooperation by the authors. However, a majority of the authors were willing to provide key reagents.

Essential revisions:

1. The Discussion makes several recommendations for increasing transparency of preclinical experiments under several headings: improving documentation and reporting; improving data, code, and materials transparency and sharing; improving preregistration of studies and analysis plans; improving rigor, reporting, and incentives with Registered Reports; and improving incentives for replication. The recommendations are presented as though none of them might have any unintended negative consequences, an unlikely scenario.

Please include a brief discussion of the possible unintended negative consequences of the recommendations.

2. In several instances, the replicators note that changes in the direction they are recommending have already been instituted by journals, which implies that this landscape is changing. It would seem prudent to suggest that an appropriately constituted committee consider the current situation before deciding what further changes might be appropriate.

Please include a brief discussion of the next steps for the preclinical cancer research community in response to your findings.

3. It is not clear how devoting more resources to the replication of experiments will help the overall process, as such experiments can only evaluate a small proportion of the published literature. In this regard, the replicators seem to have two beliefs that may be flawed.

First, they seem to believe that the scientific community will accept that the inability of replicators to replicate the findings of a particular experiment means that the initial experiment was flawed and that other investigators will be skeptical of the conclusion of the publication in question. However, that may not be the case.

Conversely, the replicators seem to believe that if other laboratories in the course of their regular studies fail to replicate the findings, either they will not publish the negative results or, if they do publish them, those results will not be accepted by the scientific community as implying that they should be skeptical of the conclusion of the initial publication. Again, this belief may be wrong.

Please comment on these two points in the manuscript.

4. Were there any particular types of experiments/techniques for which published papers lacked adequate details and/or for which authors seemed less likely to provide you with information and/or materials (such a cell lines and reagents)?

5. Please comment on how the numbers of mice you calculated were needed for the various replication experiments compares with the numbers of mice that were used in the original experiments?

Editorial points:

a) There is a tendency to represent the authors own published results in the third person (e.g. Errington and colleagues…..), giving the impression that the quoted material is from others. Please address this.

b) The manuscript cites two unpublished manuscripts:

Errington, 2021;

Errington, Mathur et al., 2021.

Please clarify how these two manuscripts are related to the present manuscript.

c) Given that a key finding of this manuscript is that many published papers do not adequately describe the methods used to obtain the results being reported, it is fitting that these authors describe what they have done in exhaustive detail! However, this can make the manuscript difficult to read in places, and most of the comments below are of an editorial nature and are intended to make the manuscript more readable.

i) The title and abstract are both too long.

ii) The passage “Replication experiments across the research life cycle” would be more readable if some of the medians and IQRs were moved to the caption for figure.

iii) The section "Design phase" would be more readable if the six sentences that start "By experiment... " were less obtrusive.

iv) There are a few passages in the discussion that unnecessarily repeat material from earlier in the article.

v) Table 1 requires a short caption to explain what are shown in columns 2, 4 and 6, and to explain why these numbers are usually different.

---

## [Author Response]

Essential revisions:1. The Discussion makes several recommendations for increasing transparency of preclinical experiments under several headings: improving documentation and reporting; improving data, code, and materials transparency and sharing; improving preregistration of studies and analysis plans; improving rigor, reporting, and incentives with Registered Reports; and improving incentives for replication. The recommendations are presented as though none of them might have any unintended negative consequences, an unlikely scenario.Please include a brief discussion of the possible unintended negative consequences of the recommendations.

We have added a brief discussion of potential unintended consequences for each of the solutions offered and pointed readers to empirical evidence when available.

2. In several instances, the replicators note that changes in the direction they are recommending have already been instituted by journals, which implies that this landscape is changing. It would seem prudent to suggest that an appropriately constituted committee consider the current situation before deciding what further changes might be appropriate.Please include a brief discussion of the next steps for the preclinical cancer research community in response to your findings.

We closed the discussion by pointing out the importance of evaluating the impact of these interventions to optimize their benefit and minimize their costs and highlighted the importance of bringing stakeholders together to evaluate the evidence in scaling up such interventions. Specifically, on Page 28 we added “an active metascience research community that evaluates the impact of these new behaviors and policies will help identify unintended negative consequences, improve their implementation, and optimize their adoption for facilitating research progress. Stakeholders in the cancer biology community including researchers, funders, societies, and institutional representatives could facilitate and support research investigations of the scientific process so that decisions about adopting these behaviors at scale can be evidence-based and clearly represent both the costs and benefits.”

3. It is not clear how devoting more resources to the replication of experiments will help the overall process, as such experiments can only evaluate a small proportion of the published literature. In this regard, the replicators seem to have two beliefs that may be flawed.First, they seem to believe that the scientific community will accept that the inability of replicators to replicate the findings of a particular experiment means that the initial experiment was flawed and that other investigators will be skeptical of the conclusion of the publication in question. However, that may not be the case.Conversely, the replicators seem to believe that if other laboratories in the course of their regular studies fail to replicate the findings, either they will not publish the negative results or, if they do publish them, those results will not be accepted by the scientific community as implying that they should be skeptical of the conclusion of the initial publication. Again, this belief may be wrong.Please comment on these two points in the manuscript.

We added discussion of these issues on pages 26-27 of the manuscript. We highlight both issues and refer readers to in-depth treatments of the role and interpretation of replication studies in advancing research.

4. Were there any particular types of experiments/techniques for which published papers lacked adequate details and/or for which authors seemed less likely to provide you with information and/or materials (such a cell lines and reagents)?

We added results, including the addition of two supplementary figures (Figure 3—figure supplement 1 and Figure 3—figure supplement 2), to page 12 of the manuscript. There was little variation in the extent of clarifications or helpfulness by category of experimental technique.

5. Please comment on how the numbers of mice you calculated were needed for the various replication experiments compares with the numbers of mice that were used in the original experiments?

We added these comparisons (replications were 25% higher in average sample size compared to original experiments) to page 12 of the manuscript.

Editorial points:a) There is a tendency to represent the authors own published results in the third person (e.g. Errington and colleagues…..), giving the impression that the quoted material is from others. Please address this.

The authorship lists between the papers are not perfectly overlapping. Using “we” is reasonable with recognition that it is a rough approximation of who contributed to the underlying research.

b) The manuscript cites two unpublished manuscripts:Errington, 2021;Errington, Mathur et al., 2021.Please clarify how these two manuscripts are related to the present manuscript.

Errington (2021) reports individual experiments that were completed as part of the Reproducibility Project: Cancer Biology but did not make it into a published Replication Study because other experiments from that Registered Report were not performed or completed.

Errington, Mathur, et al. (2021) is a meta-analysis of the statistical outcomes from the replication studies that were completed as part of the Reproducibility Project: Cancer Biology. It is now under review at *eLife* as a companion paper to this piece.

c) Given that a key finding of this manuscript is that many published papers do not adequately describe the methods used to obtain the results being reported, it is fitting that these authors describe what they have done in exhaustive detail! However, this can make the manuscript difficult to read in places, and most of the comments below are of an editorial nature and are intended to make the manuscript more readable.i) The title and abstract are both too long.

Revised and shortened. The title is now “Challenges for Assessing Reproducibility and Replicability in Preclinical Cancer Biology” and the abstract is now 200 words.

ii) The passage “Replication experiments across the research life cycle” would be more readable if some of the medians and IQRs were moved to the caption for figure.

Done.

iii) The section "Design phase" would be more readable if the six sentences that start "By experiment... " were less obtrusive.

We removed the sentences summarizing the findings by paper and retained the sentences summarizing the findings by experiment. And, we referred the reader to Figure 1—figure supplement 1 to see the data represented by paper (very similar results). We retained summary data at the papers-level in a couple of places in the manuscript when it was particularly relevant to characterize the findings for the paper as a whole.

iv) There are a few passages in the discussion that unnecessarily repeat material from earlier in the article.

We reviewed and edited the manuscript to cleanly present the findings and avoid unnecessary redundancy. We did retain the opening paragraph of the Discussion section that briefly summarizes the key findings from the results.

v) Table 1 requires a short caption to explain what are shown in columns 2, 4 and 6, and to explain why these numbers are usually different.

Done.